# Effect of vegetable oil on ovarian steroidogenesis- A transcriptome approach to understand molecular mechanisms of hypothalamus pituitary and gonad axis (HPG) in *Ompok bimaculatus*

**Sagar Chandra Mandal**[1], **Partha Sarathi Tripathy**[2,3], **Ananya Khatei**[2,4], **Ningthoujam Chaoba Devi**[1], **Pradyut Biswas**[1], **Jitendra Kumar Sundaray** [5]*, **Farhana Hoque**[5], **Janmejay Parhi**[1]

1 College of Fisheries, Central Agricultural University (Imphal), Lembucherra, Tripura, India, 2 Faculty of Biosciences and Aquaculture, Nord University, Bodø, Norway, 3 Rani Lakshmibai Central Agricultural University, Jhansi, Uttar Pradesh, India, 4 Directorate of Coldwater Fisheries Research, Bhimtal, Uttarakhand, India, 5 ICAR-Central Institute of Freshwater Aquaculture, Bhubaneswar, Odisha, India

* jsundaray@gmail.com

## Abstract

*Ompok bimaculatus* is a commercially important food fish of Northeast India. For higher production of this species, the molecular mechanisms underlying its reproductive physiology is an important aspect to study. Vegetable oils are rich in ω3 and ω6 fatty acids and fat-soluble vitamins, that enhance reproductive performances. In the present study, *O. bimaculatus* were first fed with vegetable oil at 3, 5, 7 and 9% in the diet. The best treatment was determined by the analysis of relative fecundity, fertilisation rate, Gonadosomatic Index (GSI) and Hepatosomatic Index (HIS). Comparative transcriptomics analysis of gonadal tissues between control and best treatment was done using Illumina NextSeq500. The bioinformatics analysis by Trinity v2.11.0 and edgeR v3.32.1 identified 36 DEGs related to reproduction, those were later validated by qPCR analysis. The Differentially Expressed Genes (DEGs) revealed, regulation of Hypothalamic—pituitary—gonadal axis (HPG) axis through retinoid signalling pathway, IGF pathway and folliculogenesis pathway, including inhibitory effects of dietary vegetable oil on fish gonadal maturation.

## 1. Introduction

Lipids play significant roles as sources of metabolic energy for growth and reproduction in fishes. Large amounts of lipids are required both for egg production and breeding activities [1, 2]. During maturation and spawning, there is a need for high energy in the gonads. The lipid stored in the liver and muscles is transferred and move to the gonad [3]. In adult fishes, different lipid classes in gonads, liver and muscle influence sexual maturation and spawning [4, 5]. Essential fatty acids (EFA), particularly docosahexaenoic acid (DHA: 22:6n-3) and

**Data Availability Statement:** Sequence reads are available from the NCBI sequence read archive (SRA) under the accession number PRJNA705413.

**Funding:** The authors thank the DBT-Twinning Pabda Project, Department of Biotechnology, Government of India (BT/PR25141/NER/95/1039/2017) for providing facilities and financial support to conduct the experiment. Sagar Chandra Mandal, Jitendra Kumar Sundaray, Farhana Hoque and Janmejay Parhi are the recipients of the funding award listed above. The funders had no role in study design, data collection and analysis, decision to publish, or preparation of the manuscript.

**Competing interests:** The authors have declared that no competing interests exist.

eicosapentaenoic acid (EPA: 20:5n-3), are one of the major nutritional factors significantly affecting reproductive performance in fish [6]. Phospholipids are the primary lipid required during egg yolk development, embryonic and early larval development in fishes [7]. Polyunsaturated fatty acids like arachidonic acid (AA:20:4n-6) is also considered as an essential compound because of their involvement in the production of eicosanoid [8], particularly prostaglandins, which are involved in several reproductive processes including the production of steroid hormones and gonadal development such as ovulation [9]. Most fish cannot synthesise *n-3* and *n-6* PUFA *de novo* so, they must be supplied in the diet. High unsaturated fatty acid (HUFA) and polyunsaturated fatty acids (PUFA) in brooder feed significantly affect fecundity, fertility, hatching, and viability of fish eggs and larval physiological fitness [10, 11].

*Ompok bimaculatus* (Pabda) is a highly demanded fish species in aquaculture, especially in eastern and northeastern India and Bangladesh, because of its good taste and flavor [12]. The fish has been recognized as the state fish of Tripura. Various efforts are made to improve brood stock quality to produce sufficient and good quality offspring for high production. This study aims to evaluate the effects of dietary vegetable oil inclusion on the molecular reproductive processes in Pabda. Specifically, the study investigates the reproductive genes, which are numerous and associated with processes from gonadal development to the release of milt and oocyte. The transcriptomic study focuses on the whole transcripts (RNA) transcribed from a given cell or tissue type at a particular time [13]. Comparative transcriptomics provides information on the differential expression patterns of important genes between treated and control groups. Therefore, to understand the molecular mechanisms of dietary vegetable oil supplements on the reproductive process, a comparative transcriptomics analysis was carried out. This approach identifies a set of reproductive genes responsible for reproduction and spawning using an OMICS approach and further validates these findings using qPCR.

## 2. Materials and methods

### 2.1. Ethical statement

All the fish specimens of *O. bimaculatus* used in this study were collected from the third generation of domesticated stock in the Wet Lab of College of Fisheries, CAU (Imphal), Tripura, India. Ethical approval, specimen collection and maintenance were performed strictly according to ethical standards of the Institute's Animal Ethics Committee, College of Fisheries, Central Agricultural University, Imphal, India (IAEC/COF-CAU No. A8/213422/2021). The fish were humanely euthanized using an overdose of tricaine methanesulfonate (MS-222) at a concentration of 300 mg/L, followed by decapitation to ensure death. Anesthesia was administered using tricaine methanesulfonate (MS-222) at a concentration of 100 mg/L for all handling and sampling procedures to minimize stress and discomfort. Throughout the study, efforts were made to alleviate suffering by maintaining optimal water quality, providing appropriate nutrition, and minimizing handling times. Any fish showing signs of distress or illness were promptly euthanized using the above-mentioned humane method.

### 2.2. Experimental setup

The third generation of the domesticated stock of *O. bimaculatus* was kept in circular tanks (12 feet diameter, 4 feet depth, 2000 litre water) for six months in triplicates, i.e., three tanks each in control and treatment groups. The stocking density was kept at 30 fish per tank, and proper water quality parameters were maintained to minimise environmental effects on the experiment. The control group were fed with an artificially formulated diet without adding any vegetable oil (Table 1). The treated group were fed a diet with the inclusion of vegetable oil at 3, 5, 7 and 9% for T1, T2, T3 and T4, respectively (Table 1). The vegetable oil used was

**Table 1. Formulation and composition of the experimental diets (in % per kg).**

| Ingredients | Control | T1 | T2 | T3 | T4 |
|---|---|---|---|---|---|
| Fishmeal[1] | 20 | 20 | 20 | 20 | 20 |
| Soyabean[1] | 08 | 08 | 08 | 08 | 08 |
| Wheat bran[1] | 11 | 11 | 11 | 11 | 11 |
| Casein[2] | 18 | 18 | 18 | 18 | 18 |
| Gelatine[2] | 5 | 5 | 5 | 5 | 5 |
| White dextrin [2] | 14 | 14 | 14 | 14 | 14 |
| Cellulose[2] | 17.48 | 14.48 | 12.47 | 10.46 | 8.46 |
| Vegetable oil [3] | 00 | 03 | 05 | 07 | 09 |
| Vitamin Premix (Vit-E free)[4] | 02 | 02 | 02 | 02 | 02 |
| Mineral Premix[4] | 03 | 03 | 03 | 03 | 03 |
| BHT[2] | 0.02 | 0.02 | 0.02 | 0.02 | 0.02 |
| Carboxymethyl cellulose [2] | 1.5 | 1.5 | 1.5 | 1.5 | 1.5 |

[1]Obtained from College of Fisheries, CAU, Lembucherra, Agartala, India.

[2]Hi-media Laboratories, Mumbai, India.

[3]Ruchi Soya Pvt. Ltd., Raighad, India.

[4]Vitamin- Mineral Premix (per kg of diet): vitamin A, 2000 IU; vitamin B1 (thiamin), 5 mg; vitamin B2 (riboflavin), 5 mg; vitamin B6, 5 mg; vitamin B12, 0.025 mg; vitamin D3, 1200 IU; vitamin K3, 2.5 mg; folic acid, 1.3 mg; biotin, 0.05 mg; pantothenic acid; calcium, 20 mg; inositol, 60 mg; ascorbic acid (35%), 110 mg; niacinamide, 25 mg; MnSO4, 10 mg; MgSO4, 10 mg; KCl, 95 mg; NaCl, 165 mg; ZnSO4, 20 mg; KI, 1 mg; CuSO4, 12.5 mg; FeSO4, 105 mg; Na2SeO3, 0.1 mg; Co, 1.5 mg.

produced by Ruchi Soya Pvt. Ltd., Raigad, India, that primarily consists of 90–95% refined soybean oil, enriched with 1–2% natural antioxidants such as Vitamin E, and 0.1–0.5% anti-foaming agents like Dimethylpolysiloxane. It also contains trace amounts of additives, including emulsifiers and stabilizers. Nutritionally, per 100g, the oil provides 884 kcal of energy and contains 100g of total fat, comprising 15g of saturated fat, 23g of monounsaturated fat, and 62g of polyunsaturated fat. The oil has zero trans fat and cholesterol, and it supplies 12mg of Vitamin E, meeting 80% of the daily value. The crude protein content was 30–35% in each feed. The best treatment among these, i.e., T2, was taken for transcriptomics analysis. Moreover, the relative fecundity, fertilisation rate, GSI (Gonado Somatic Index), HSI (Hepato Somatic Index), etc., were calculated based on the methods of earlier published work [14]. These parameters were calculated using ten fish specimens from each replicate.

## 2.3. Sample collection and RNA extraction

The ovary tissues were collected in triplicates from each group (control and T2) and kept at -20˚C for qPCR analysis. The tissue samples were pooled from the control and treated group (T2) and were marked as PCO (Pabda Control Ovary) and PTO (Pabda Treated Ovary), respectively, for transcriptomics analysis. The tissue samples were kept in liquid nitrogen until RNA isolation. The total RNA was isolated by RNeasy Plus Universal Mini Kit (Qiagen, USA), as per the manufacturer's protocol. The quality and quantity of the isolated RNA were checked using 1% denaturing RNA agarose gel and NanoDrop, respectively.

## 2.4. Library preparation and sequencing

Library preparation was performed with Illumina TruSeq Stranded mRNA Sample Prep kit (Illumina, USA) according to the standard protocol set from the manufacturer. The RNA

sample concentration used from both samples were 1355 ng/μl. The library quality was tested using Agilent 4200 Tape Station system (Agilent Technologies, USA). The Paired-end (PE) Illumina libraries were loaded onto NextSeq500 for cluster generation and sequencing using 2 x 75 bp chemistry. This PE sequencing allows the template fragments to be sequenced in both forward and reverse directions. The illumine kit reagents were used in the binding of samples to complementary adapter oligos on the PE flow cell. The adapters were designed to allow selective cleavage of the forward strands after re-synthesis of the reverse strand during sequencing. The copied reverse strand was used to sequence from the opposite end of the fragment.

## 2.5. *de novo* assembly and DE analysis

The raw reads generated in fastq format were then taken for quality checking by FastQC v0.11.9 [15]. The low-quality reads (phred score < 30), Illumina adapters and any required overrepresented sequences for each sample were removed by quality trimming using Cutadapt v3.2 [16]. Sequence reads are available from the NCBI sequence read archive (SRA) under the accession number PRJNA705413. Trimmed reads were again checked for quality using FastQC v0.11.9 [15]. The final quality reads of all the samples were concatenated to a single file and loaded onto Trinity software package v2.11.0 [17] with default parameters to assemble high-quality reads into contigs. The Trinity package helped to reconstruct the transcripts and isoforms into their final form. The transcriptome assembly quality was checked using sanger-pathogens/assembly-stats v1.0.1 to generate standard metrics and N50 statistics. Mapping and abundance estimation for each sample were performed using the align_and_estimate_abundance.pl utility in Trinity [17] with default parameters, employing RSEM as the method for abundance estimation. The count tables generated for each sample were merged for further differential expression (DE) analysis using Trinity's abundance_estimates_to_matrix.pl utility [17]. The merged RSEM isoform count matrix was also used for calculating fold changes between PCO and PTO by edgeR v3.32.1 [18] package in R. Additionally, Fragments Per Kilobase of Transcript per Million Mapped (FPKM) values for each sample were calculated using the run_TMM_normalization_write_FPKM_matrix.pl function in Trinity [17]. The percentage of reads assembled was computed using bowtie2 v2.4.2 [19], and the completeness of the assembly was assessed using BUSCO v5.0.0 [20] with the "Actinopterygii" lineage dataset (actinopterygii_odb10, 2021-02-19). BUSCO was run in transcriptome assembly mode.

## 2.6. Functional annotation and CDS prediction of differentially expressed transcript

The coding transcripts from the *de novo* assembled transcripts were detected using TransDecoder v5.5.0. Then the functional annotation of the CDS was performed using DIAMOND (BLASTX alignment mode) v2.0.7 [21] to find the homologous sequences for the CDS against NR (non-redundant protein database) from NCBI. Reads of PCO and PTO were pooled to identify the CDS of each sample, and the reads from each of the samples were mapped on the final set of pooled CDS using BWA-MEM toolkit v0.7.17 [22]. The read count (RC) values were calculated from the resulting mapping, and those CDS with at least 85% coverage and 3X read depth were considered for downstream analysis for each sample. These FPKM values of the top 50 differentially expressed genes (DEGs) and reproductive genes were used for heatmap generation between PCO and PTO using gdata v2.18.0 [23] and pheatmap v1.0.12 [24] package in R. Among the top 50 DEGs, 25 were taken as up-regulated and 25 as down-regulated. Log2 Fold Change was calculated for further analysis of differential expression of each transcript between PCO and PTO by taking the log of PTO FPKM vs PCO FPKM value. All

the DEGs with p-value <0.05 and log2 fold changes were taken for volcano plot generation in gdata v2.18.0 [23] and gplots v3.1.1 [25] package in R.

## 2.7. Gene ontology (GO) and KEGG pathway mapping

The DEGs were taken for GO and KEGG pathway determination on shinyGO v0.61 [26], taking a p-value cut-off of 0.05. All the genes were analysed against three GO categories, i.e., Biological Process, Cellular Component and Molecular Function. The output of shinyGO analysis was plotted in the WEGO plot [27].

## 2.8. qPCR validation

The QuantStudio™ 5 Real-Time PCR (Applied Biosystems, USA), based on SYBR green chemistry, was used to validate candidate genes related to reproduction. The gene-specific primers were designed based on transcriptomics data. The gonad tissues were sampled from both PCO and PTO in triplicates. Total RNA was isolated from all the tissues, using Quick-RNA Miniprep Plus Kit (Zymo Research, USA) and cDNA was prepared, using iScript™ cDNA Synthesis Kit (Bio-Rad, USA) as per the manufacturer's protocol. qPCR reaction was performed by adding 5 μl of SYBR green master mix (Thermo Scientific, USA), forward and reverse gene-specific primers 1 μl each (1 pmol/μl), 1 μl of template cDNA (1 μg/μl) and nuclease-free water to make it up to a total volume of 10 μl. The. β-actin was taken as an internal control for expression level normalisation. Three technical replicates were made for each sample during qPCR. The analysis was done by the method of Pfaffl [28].

## 2.9. Statistical analysis

Differentially expressed genes (DEGs) were identified, and statistical significance (p-value <0.05) was determined. Heatmaps and volcano plots of DEGs were created using gdata v2.18.0, pheatmap v1.0.12, and gplots v3.1.1 packages in R. Gene ontology (GO) and KEGG pathway analysis were performed using shinyGO v0.61 with a p-value cut-off of 0.05. The qPCR validation involved analyzing gene expression in triplicates with QuantStudio™ 5 Real-Time PCR. The statistical significance was assessed, and results were reported as mean ± standard error.

# 3. Results

## 3.1. Wet-lab experiment

The fertilisation rate and GSI were found to be highest for the T2 group of fishes i.e. averaging 80.02 and 23.83, respectively (Table 2). Interestingly, the HSI was found to be the lowest for T2 i.e. averaging 0.65 (Table 2). The relative fecundity was found to be highest for T2 and T3 groups, averaging 756.66 eggs/g body weight. From all the calculations, the T2 group was the best treatment, and no mortality was seen in any experimental group except T3. The T2 was identified as the best treatment for transcriptomics analysis based on several parameters observed in Table 2. The relative fecundity of T2 was significantly higher (743.39±2.50, 778.12 ±2.44, 749.16±3.6 eggs/g body weight) compared to other treatments and the control group. Additionally, T2 showed an improved fertilization rate (81.07±1.93, 79.00±2.06, 79.99±1.83%) and a notably higher Gonado Somatic Index (GSI) (24.46±1.07, 23.80±0.02, 23.25±0.40), indicating better reproductive health and development.

**Table 2. Different parameters of female *O. bimaculatus* brood fed with diets containing varying levels of lipid.** Three biological replicates (R1, R2 and R3) in each treatment were taken. In each biological replicate, ten fish specimens were analysed. The values are represented as mean ± Standard Error (S.E.).

| Parameters | CR1 | CR2 | CR3 | T1R1 | T1R2 | T1R3 | T2R1 | T2R2 | T2R3 | T3R1 | T3R2 | T3R3 | T4R1 | T4R2 | T4R3 |
|---|---|---|---|---|---|---|---|---|---|---|---|---|---|---|---|
| Relative Fecundity (Eggs/g body weight) | 399.55 ±5.39 | 385.12 ±1.3 | 345.21 ±2.12 | 486.19 ±4.42 | 456.56 ±4.5 | 477.23 ±1.33 | 743.39 ±2.50 | 778.12 ±2.44 | 749.16 ±3.6 | 754.46 ±4.31 | 789 ±5.6 | 753.45 ±4.21 | 610.47 ±4.28 | 612.21 ±5.2 | 608 ±4.55 |
| Weight of each egg (mg) | 0.08 ±0.00 | 0.07 ±0.00 | 0.07 ±0.00 | 0.07 ±0.00 | 0.07 ±0.00 | 0.06 ±0.00 | 0.08 ±0.00 | 0.08 ±0.00 | 0.07 ±0.00 | 0.08 ±0.00 | 0.08 ±0.00 | 0.07 ±0.00 | 0.07 ±0.00 | 0.06 ±0.00 | 0.07 ±0.00 |
| Volume of Eggs/gm of fish | 1.13 ±0.04 | 1.11 ±0.01 | 1.09 ±0.05 | 1.74 ±0.05 | 1.68 ±0.02 | 1.66 ±0.04 | 2.08 ±0.05 | 2.05 ±0.03 | 1.99 ±0.02 | 1.82 ±0.04 | 1.72 ±0.01 | 1.79 ±0.02 | 1.75 ±0.07 | 1.79 ±0.04 | 1.70 ±0.06 |
| Fertilization rate | 51.27 ±2.08 | 48.07 ±1.08 | 50.27 ±1.12 | 57.03 ±4.56 | 53.15 ±1.05 | 56.21 ±2.01 | 81.07 ±1.93 | 79.00 ±2.06 | 79.99 ±1.83 | 77.29 ±1.33 | 75.20 ±1.09 | 72.27 ±2.10 | 62.84 ±2.03 | 58.02 ±1.04 | 59.30 ±1.44 |
| GSI (Gonado somatic Index) | 10.70 ±0.61 | 10.12 ±0.5 | 10.50 ±0.23 | 12.82 ±0.56 | 12.05 ±0.45 | 12.02 ±0.7 | 24.46 ±1.07 | 23.80 ±0.02 | 23.25 ±0.40 | 21.84 ±0.39 | 21.20 ±0.03 | 21.50 ±0.61 | 16.06 ±1.20 | 16.26 ±0.5 | 16.98 ±0.26 |
| HSI (Hepato Somatic Index) | 0.96 ±0.03 | 0.97 ±0.05 | 0.87 ±0.02 | 0.80 ±0.03 | 0.76 ±0.01 | 0.78 ±0.02 | 0.65 ±0.04 | 0.63 ±0.03 | 0.65 ±0.02 | 0.67 ±0.01 | 0.66 ±0.03 | 0.64 ±0.05 | 0.74 ±0.03 | 0.74 ±0.01 | 0.68 ±0.04 |

## 3.2. RNA-Seq data analysis

The data generated for PCO and PTO were found to be 6.37 and 4.56 Gb, respectively. The total number of reads for PCO and PTO were found to be 42,479,658 and 30,366,474, respectively. The Trinity analysis generated 150,836 pooled transcripts with maximum and minimum transcript lengths of 16,683 and 201, respectively. The completeness of the BUSCO analysis on the present transcriptomic dataset was found to be 81%. Out of the total 2950 complete BUSCOs (C), 2488 complete and single-copy BUSCOs (S) and 462 complete and duplicated BUSCOs (D) were found. Moreover, 75 Fragmented BUSCOs (F) and 615 missing BUSCOs (M) were found in 3640 total BUSCO groups searched from the actinopterygii_odb10 dataset. The GC content was 47.24%. In total, 99.28% of the reads were assembled.

The present transcriptomics showed maximum BLAST hits with *Pangasianodon hypophthalamus*, *Ictalurus punctatus* and *Tachysurus fulvidraco*. The total number of CDS for PCO and PTO were found to be 18,421 and 17,830, respectively. The volcano plot for all the DEGs in the dataset has been shown in Fig 1. The heatmap for the top 50 DEGs has been shown in Fig 2. The GO analysis for the DEGs (WEGO plot) has been shown for PCO and PTO in Fig 3a and 3b, respectively. We found candidate genes related to reproduction among

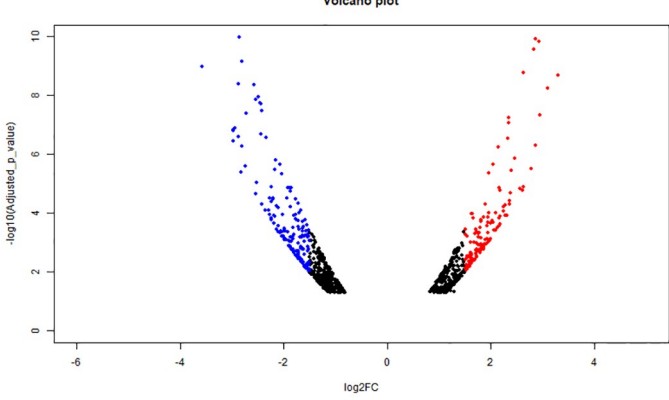

**Fig 1. Volcano plot showing top up and down-regulated genes.** Up-regulated and down-regulated genes are marked with red and blue dots, respectively. The logFC and -log$_{10}$ adjusted p-values were plotted using R.

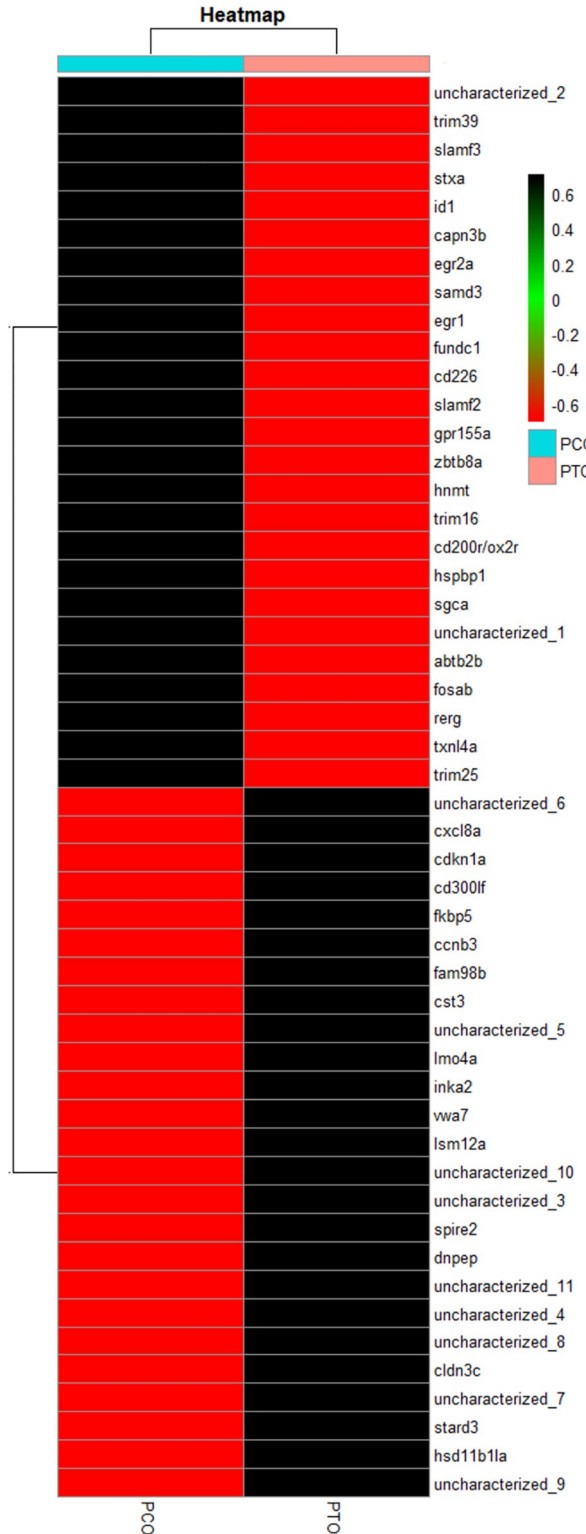

**Fig 2. Heatmap showing top 50 up and down-regulated genes.** FPKM values of the top 50 DEGs and reproductive genes were used for heatmap generation between PCO and PTO using R.

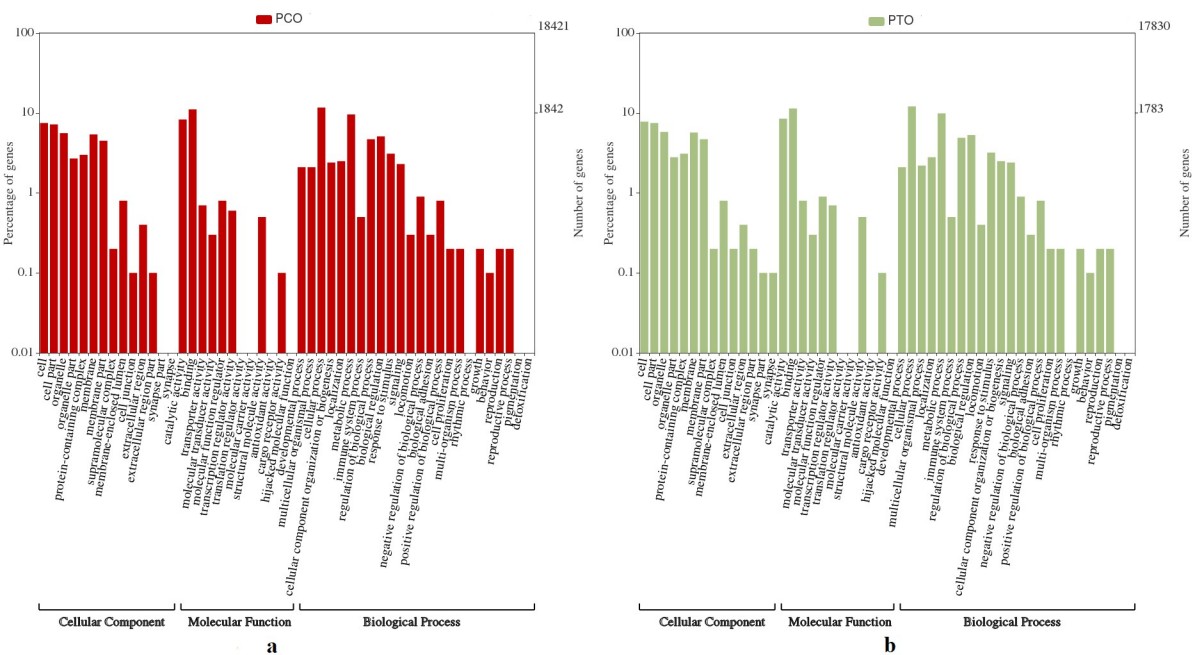

**Fig 3.** Gene ontology enrichment analysis of DEGs for **a.** PCO and **b.** PTO. DEGs described were analyzed for enrichment in three GO categories i.e. biological process, cellular component, and molecular function.

the top 50 DEGs. But in addition to this we found other genes related to reproduction in the transcriptomics data, which were differentially expressed.

### 3.3. qPCR analysis

The qPCR validation of candidate genes related to reproduction supported the differential expression patterns analysed in the transcriptomics data. The *stard3*, *SF1-A*, *cyp19a1a*, *rxrba*, *rxrbb*, *rxrga*, *egr2a*, *cyp17a1*, *hsd3b*, etc. were differentially expressed in the gonad tissues. The *aqp3a* and *cyp17a1* were found to be the top down-regulated gene. Similarly, *hsd3b*, *ar* and *hsp70.1* were found to be the top up-regulated gene in T2. The average of technical replicates for each biological replicates have been shown in the present study. The differential expressions are shown as log2 fold changes than control in Table 3.

### 4. Discussion

The present study investigates the molecular effects of vegetable oil on the female reproductive process in *O. bimaculatus*. Previous research has highlighted the positive impact of dietary lipids on ovarian development [29–32]. The vegetable oils (Ruchi Soya Pvt. Ltd., Raigad, India), which are rich in ω3 and ω6 fatty acids and fortified with vitamins A, D, E, and K, offer substantial nutritional benefits. In this study, reproductive indices demonstrated significant improvements in fish treated with vegetable oil compared to the control group. The Gonado Somatic Index (GSI) was markedly higher in the treated group, indicating enhanced ovarian development facilitated by the components of vegetable oil. Furthermore, the relative fecundity was significantly elevated in the treated group, suggesting that vegetable oil supplementation effectively increased the number and quality of oocytes.

**Table 3. Differential expression of candidate genes related to reproduction in PTO (T2 group of fishes).** The transcriptomics data are shown as log2 fold changes than PCO with their respective p-values. Similarly, qPCR values are shown as log2 fold changes ± S.E. than PCO in three biological replicates, i.e., T2R1, T2R2 and T2R3.

| Gene name | Gene symbol | Transcriptomics | | qPCR | | |
|---|---|---|---|---|---|---|
| **Down-regulated** | | | | | | |
| | | **log2 Fold Change** | **p-value** | **log2 Fold Change ± S.E.** | | |
| | | | | **T2R1** | **T2R2** | **T2R3** |
| Spermatid perinuclear RNA-binding protein | strbp-002 | -1.86 | 0.002187 | 0.47±0.04 | 0.41±0.07 | 0.36±0.05 |
| Estrogen receptor | esr1 | -2.20 | 0.002189 | 0.42±0.08 | 0.49±0.06 | 0.41±0.02 |
| Steroid 17-alpha-hydroxylase/17,20 lyase | cyp17a1 | -1.46 | 1.00E-05 | 0.19±0.03 | 0.18±0.04 | 0.23±0.03 |
| Aquaporin-4 | aqp4 | -2.16 | 0.003044 | 0.66±0.04 | 0.63±0.06 | 0.66±0.03 |
| Estradiol 17-beta-dehydrogenase 1 | hsd17b1 | -2.17 | 0.002953 | 0.48±0.04 | 0.48±0.06 | 0.51±0.06 |
| Membrane-associated progesterone receptor component 1 | pgrmc1 | -2.14 | 7.54E-05 | 0.76±0.03 | 0.79±0.04 | 0.74±0.07 |
| Aquaporin-3a | aqp3a | -2.13 | 0.003342 | 0.16±0.03 | 0.17±0.02 | 0.21±0.05 |
| Retinoic acid receptor alpha | raraa | -2.13 | 0.004196 | 0.53±0.08 | 0.52±0.07 | 0.52±0.08 |
| Follicle-stimulating hormone receptor | fshr | -2.10 | 1.65E-07 | 0.64±0.04 | 0.69±0.05 | 0.63±0.02 |
| Bromodomain testis-specific protein | brdt | -2.09 | 7.30E-07 | 0.44±0.06 | 0.46±0.05 | 0.49±0.03 |
| Retinoic acid receptor RXR-gamma-B | rxrgb | -2.08 | 0.004081 | 0.48±0.07 | 0.47±0.04 | 0.44±0.05 |
| Lutropin-choriogonadotropic hormone receptor | lhcgr | -2.07 | 9.13E-06 | 0.39±0.07 | 0.39±0.04 | 0.37±0.03 |
| Follistatin | fsta | -1.90 | 9.22E-06 | 0.33±0.06 | 0.31±0.04 | 0.31±0.03 |
| **Up-regulated** | | | | | | |
| **Gene name** | **Gene symbol** | **Transcriptomics** | | **qPCR** | | |
| | | | | **log2 Fold Change ± S.E.** | | |
| | | **log2 Fold Change** | **p-value** | **T2R1** | **T2R2** | **T2R3** |
| Thyroid hormone receptor-associated protein 3 | thrap3b | 2.02 | 0.002151 | 3.47±0.07 | 3.49±0.09 | 3.55±0.11 |
| Cytochrome P450 | cyp19a1a | 2.10 | 1.47E-05 | 2.42±0.08 | 2.39±0.03 | 2.48±0.09 |
| Spermatid perinuclear RNA-binding protein | strbp | 2.11 | 0.003563 | 3.23±0.03 | 3.19±0.04 | 3.18±0.07 |
| Aquaporin FA-CHIP-like | aqp7 | 2.13 | 0.003023 | 3.69±0.04 | 3.66±0.06 | 3.74±0.04 |
| Retinoic acid receptor RXR-beta-B | rxrbb | 2.13 | 0.004023 | 1.48±0.06 | 1.44±0.05 | 1.41±0.03 |
| Steroidogenic factor 1 | nr5a1a | 2.20 | 0.003773 | 2.44±0.07 | 2.44±0.11 | 2.41±0.13 |
| Retinoic acid receptor RXR-beta-A | rxrba | 1.33 | 0.002826 | 1.41±0.04 | 1.47±0.05 | 1.47±0.07 |
| 3 beta-hydroxysteroid dehydrogenase/Delta 5—4-isomerase | hsd3b | 2.14 | 9.04E-05 | 4.24±0.07 | 4.16±0.11 | 4.25±0.09 |
| Membrane-associated progesterone receptor component 2 | pgrmc2 | 1.94 | 2.90E-05 | 2.19±0.04 | 2.19±0.03 | 2.21±0.05 |
| Androgen receptor | ar | 1.53 | 0.00358 | 4.43±0.13 | 4.39±0.09 | 4.39±0.05 |
| Progesterone receptor | pgr | 2.11 | 1.89E-05 | 3.21±0.05 | 3.27±0.04 | 3.23±0.06 |
| Estradiol 17-beta-dehydrogenase 2 | hsd17b2 | 1.84 | 0.002951 | 4.49±0.09 | 4.44±0.07 | 4.44±0.03 |
| D(1A) dopamine receptor | drd1a | 1.88 | 7.58E-07 | 2.49±0.08 | 2.43±0.05 | 2.47±0.04 |
| Retinoic acid receptor RXR-gamma-A | rxrga | 1.92 | 0.001885 | 1.42±0.08 | 1.42±0.08 | 1.46±0.09 |
| Epidermal growth factor | egf | 2.09 | 0.002826 | 3.34±0.11 | 3.36±0.07 | 3.34±0.03 |
| Transforming growth factor alpha | tgfa | 1.89 | 9.04E-05 | 2.66±0.06 | 2.73±0.09 | 2.63±0.03 |
| Insulin like growth factor 1 | igf1 | 1.97 | 2.90E-05 | 1.88±0.05 | 1.89±0.04 | 1.89±0.05 |
| Insulin like growth factor 2a | igf2a | 2.14 | 0.00358 | 3.47±0.05 | 3.52±0.07 | 3.42±0.09 |
| Insulin like growth factor 3 | igf3 | 2.02 | 0.003773 | 1.56±0.04 | 1.49±0.05 | 1.56±0.07 |
| Nuclear receptor subfamily 5, group A, member 1a/ steroidogenic factor 1 | SF1-A | 2.15 | 0.002826 | 3.33±0.07 | 3.29±0.03 | 3.28±0.05 |
| Heat shock cognate 70-kd protein, tandem duplicate 1 | hsp70.1 | 2.11 | 2.90E-05 | 4.23±0.09 | 4.14±0.11 | 4.17±0.11 |
| Star-related lipid transfer (START) domain containing 3 | stard3 | 2.28 | 0.000122 | 3.12±0.06 | 3.07±0.04 | 3.15±0.11 |
| Retinoic acid receptor RXR-gamma-B | rxrgb | 2.08 | 0.004081 | 1.46±0.06 | 1.48±0.05 | 1.41±0.09 |

Another important parameter measured in this study is the Hepatosomatic Index (HSI), which reflects ovarian status by indicating the developmental process of oocytes, as vitellogenesis occurs in the liver. In most teleost, the HSI is higher during the initial ripening of the ovary and decreases toward final maturation [33]. This pattern occurs because the liver is heavily involved in vitellogenesis during the initial phase, with most stored energy being diverted toward ovarian development as the process progresses [33]. A significant decrease in HSI was observed in the treated group, suggesting more efficient ovarian development.

Comparative transcriptomics was conducted to analyze the molecular differences between fish fed a regular diet and those fed a vegetable oil-enriched diet. This approach allowed for the identification of several essential genes involved in ovarian development and vitellogenesis. These genes play critical roles in processes such as hormone regulation, oocyte maturation, and energy allocation for reproductive activities. The transcriptomic analysis revealed distinct differential expression patterns of these genes in fish fed the vegetable oil-enriched diet compared to those on the regular diet. To validate these findings, qPCR analysis was performed on the same genes, confirming the differential expression patterns observed in the NGS data. The consistency between the qPCR results and the NGS data strengthens the reliability of the transcriptomic findings. The regulation of these genes aligned with the improved reproductive indices observed in the vegetable oil-treated group, such as higher GSI and increased relative fecundity. These molecular insights provide a comprehensive understanding of how vegetable oil supplementation enhances ovarian development and overall reproductive performance in *O. bimaculatus*. This study underscores the potential of dietary interventions in aquaculture to improve broodstock quality and reproductive success.

STAR gene (*stard3*) is the initiator of ovarian steroidogenesis in fish [34]. The star protein transports cholesterol to the inner mitochondrial membrane, where steroidogenesis begins with converting cholesterol to pregnelone. The *stard3* gene was upregulated in the fishes fed with vegetable oil compared to the control group of fishes. This gene is also among the top 50 differentially expressed genes in the present transcriptomics data. Vegetable oils contain a higher proportion of linoleic acid, which enhances steroidogenesis in vertebrates [35]. Vegetable oil contains a high amount of palmitic acid precursor of palmitoylethanolamide (PEA) [36]. This PEA is responsible for the synthesis of star protein. This might be the reason for the increase in the production of star protein. Another essential gene that regulates *stard3* is the steroidogenic factor 1 (*SF1-A*). This gene was also found to be upregulated in the treatment group of fish.

Vegetable oils have a higher amount of linoleic acid that enhances ovarian physiology. In the present study, the *cyp19a1* gene was found to be upregulated. In some fishes, down-regulation of this gene disrupts the ovarian process and leads to testicular development [37]. The *cyp19a1* shows stage-specific expressions in zebrafish and has been found abundantly in the follicular layer of vitellogenic oocytes [38]. This suggests that *cyp19a1* plays a significant role in vitellogenesis and its upregulation depicts its role in enhancing the vitellogenesis process.

Genes involved in the retinoid signalling pathways, specifically *rxrba*, *rxrbb*, and *rxrga*, exhibited up-regulation in the treatment group of fish. In females of higher vertebrates, retinoic acid signalling plays a major role in folliculognesis and affects the number and quality of oocytes [39]. Vegetable oils are fortified with vitamin A, which is the source of retinoic acid. This retinoic acid enhances the retinoid signalling pathway, thereby improving the female reproductive performance. It has been found in previous studies that *hsd17b*, which is an essential factor in ovarian steroidogenesis, is positively regulated through the retinoic acid signalling pathway [40]. Higher production of *hsd17b2* is induced by retinoic acid. In the present analysis, *hsd17b2* was up-regulated. This gene is responsible for the interconversion of Estrone (E1) to Estradiol (E2) [41]. In the present study, 17β-estradiol receptor (*esr1*) was down-

regulated, suggesting that E1 is getting converted to E2. E2 is the main regulatory factor for vitellogenesis and the final maturation of the oocyte. Although E2 is produced by the conversion of both testosterone and E1, the major pathway traversed its production is the E1 signalling pathway rather than the testosterone [42]. The E2 binds to globulin in the blood and traverses to the liver for vitellogenesis. The down-regulation of *esr1* in the treated fishes also suggests that, its expression is stage and site specific in ovary.

The classical pathway in steroidogenesis involves the breakdown of cholesterol into pregnelone, after which the hydroxylation of pregnelone results in the formation of 17α-OH-progesterone. This involves *cyp17a1* and *hsd3b*, both of which were up-regulated in the present study. The conversion into 17α-OH-progesterone into 17α-20β-dihydroxy-4-pregnen-3-one requires *hsd20b*, which was found to be up-regulated in the treatment group of fishes. Other vital genes involved in enhancing oocyte maturation, such as *egf* and *tgfa*, were also found up-regulated in the treatment group of fishes, which suggests that vegetable oil enhances the process of vitellogenesis and accelerates final oocyte maturation [43].

Interestingly, the cyclin-dependent kinase family genes mainly were found up-regulated in the present study. The highly up-regulated *cdk2*, also known as active MPF (maturation promoting factor), indicates that the incorporation of vegetable oil enhanced the maturation process of the ovary. The CDKs are primarily associated with cell differentiation and proliferation, which also holds true for germ cells [44]. Since the oocytes are in developing stages, CDKs (*cdk1*, *cdk2*, *cdk13*, *cdk15* and *cdk16*) were up-regulated in the present study.

During the folliculogenesis of the oocytes, testosterone and 11-ketotestosterone, which are primarily associated with male steroidogenesis, also have important roles. These two androgens express in a stage-specific manner in the ovary and stimulate cortical alveoli production [45]. Therefore, androgen receptors show increased expression in the ovary during folliculogenesis. It has been studied that androgen receptor antagonist, when administered, induces early follicular atresia and inhibits growth responses in oocytes [46]. The progesterone receptor was found to be up-regulated in the present experiment. This indicates that increased intake of dietary lipids enhanced the expression of both androgen and progesterone receptors, thereby aiding ovarian development.

A higher amount of linoleic acid in the vegetable oils enhanced ovarian steroidogenesis. Treating with linoleic acid increases the expression of insulin-like growth factors (IGFs) in higher vertebrates [47]. In the present study, the insulin-like growth factors (1,2 and 3) were also up-regulated significantly. The *igf1* and *igf2* are found in the liver and gonads of vertebrates. However, *igf3* is known to be found only in fish and is gonad-specific. The *igf1* promotes oocyte maturation and proliferation, whereas *igf2* has a more specific role in folliculogenesis and exhibits germinal vesicle breakdown (GVBD). The *igf3* also has similar functions in teleosts. In some fishes like the killifish, it has been found that IGFs are more potent in stimulating oocyte maturation than 17a-20b-dihydroxy-4-pregnen-3-one (MIS) [48].

In the present study, specific genes responsible for the inhibition of oocyte maturation were significantly downregulated. Among these genes, the follistatin group (*fsta*) was notably reduced in expression. Follistatin is a critical regulatory protein in the epidermal growth factor (*egf*) and transforming growth factor-alpha (*tgfa*) pathways [49]. The *egf* is known to accelerate oocyte maturation, while follistatin can inhibit this process by blocking the EGF pathway, thus acting as an oocyte maturation inhibitor. The significant down-regulation of the *fsta* gene in the treated group of fish suggests that the incorporation of vegetable oil in the diet enhances the factors responsible for promoting oocyte maturation while reducing the inhibitory factors. This dietary intervention appears to create a more favourable molecular environment for oocyte development by reducing the expression of inhibitors like follistatin. Consequently, the

treated group experiences an up-regulation of pathways that support oocyte maturation. Additionally, while *tgfa* is known to facilitate oocyte maturation, its counterpart, transforming growth factor-beta (*tgfb*), inhibits this process [50]. The findings indicate that vegetable oil supplementation enhances the positive regulators (such as *tgfa* and *egf*) of oocyte maturation and simultaneously down-regulates negative regulators (such as follistatin and potentially *tgfb*). This dual action likely contributes to the observed improvements in ovarian development and reproductive indices in the treated group of fish.

Another important gene related to reproduction is the *hsp70* binding protein. Since *hsp70* is found in the gonads, it can not be ignored that it modulates reproductive stress. In higher vertebrates, it has been found that *igf1* and *fsh* can reduce the expression of *hsp70* and reduce thermal and nutritional stress effects on the ovary [51], which might be the reason behind the down-regulation of *hsp70* in the treatment group of fish. Since the expression of both *igf1* and *hsp70* were significantly higher in the present study, it can be stated that there is little or no reproductive stress. Similarly, RAS-like estrogen-regulated growth inhibitor (*rerg*) that inhibits cellular proliferation was significantly down-regulated in the treatment group of fish. There is a significant down-regulation of the EGR group of genes (*egr1* and *egr2a*). These genes are highly expressed during stress, apoptotic signalling and tissue damage [52, 53].

The above findings suggest that incorporating fortified vegetable oils can prove highly beneficial for oocyte maturation and ovarian development. It improves ovarian steroidogenesis at a molecular level and maintains ovarian physiological processes.

## Acknowledgments

The authors are thankful to the Vice Chancellor, Central Agricultural University, Imphal, India, for providing infrastructure facilities to carry out this research.

## Author Contributions

**Conceptualization:** Sagar Chandra Mandal, Partha Sarathi Tripathy, Ningthoujam Chaoba Devi, Pradyut Biswas, Jitendra Kumar Sundaray, Farhana Hoque, Janmejay Parhi.

**Data curation:** Partha Sarathi Tripathy, Jitendra Kumar Sundaray, Farhana Hoque, Janmejay Parhi.

**Formal analysis:** Sagar Chandra Mandal, Ananya Khatei, Ningthoujam Chaoba Devi, Pradyut Biswas, Jitendra Kumar Sundaray, Janmejay Parhi.

**Funding acquisition:** Sagar Chandra Mandal, Jitendra Kumar Sundaray.

**Investigation:** Sagar Chandra Mandal, Partha Sarathi Tripathy, Ananya Khatei, Ningthoujam Chaoba Devi, Pradyut Biswas, Farhana Hoque.

**Methodology:** Sagar Chandra Mandal, Partha Sarathi Tripathy, Ananya Khatei, Ningthoujam Chaoba Devi, Pradyut Biswas, Jitendra Kumar Sundaray.

**Project administration:** Sagar Chandra Mandal, Jitendra Kumar Sundaray.

**Resources:** Sagar Chandra Mandal, Pradyut Biswas, Jitendra Kumar Sundaray.

**Software:** Partha Sarathi Tripathy, Pradyut Biswas, Farhana Hoque.

**Supervision:** Sagar Chandra Mandal, Janmejay Parhi.

**Validation:** Partha Sarathi Tripathy, Ananya Khatei, Ningthoujam Chaoba Devi, Janmejay Parhi.

**Visualization:** Sagar Chandra Mandal, Partha Sarathi Tripathy, Ningthoujam Chaoba Devi, Janmejay Parhi.

**Writing – original draft:** Partha Sarathi Tripathy, Ningthoujam Chaoba Devi.

**Writing – review & editing:** Sagar Chandra Mandal, Ananya Khatei, Jitendra Kumar Sundaray, Janmejay Parhi.

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
