## [Decision Letter · Decision Letter 0]

5 Jul 2024

PONE-D-24-14273Effect of vegetable oil on ovarian steroidogenesis- A transcriptome approach to understand molecular mechanisms of hypothalamus pituitary and gonad axis in Ompok bimaculatusPLOS ONE

Dear Dr. Sundaray,

Thank you for submitting your manuscript to PLOS ONE. After careful consideration, we feel that it has merit but does not fully meet PLOS ONE’s publication criteria as it currently stands. Therefore, we invite you to submit a revised version of the manuscript that addresses the points raised during the review process.

 Specific comments by the academic editor are included in the attached filesThe quality of the figures need to be modified.

We look forward to receiving your revised manuscript.

Kind regards,

Amel Mohamed El Asely

Academic Editor

PLOS ONE

Journal Requirements:

https://journals.plos.org/plosone/s/file?id=ba62/PLOSOne_formatting_sample_title_authors_affiliations.pdf"

3. Thank you for your submission to PLOS ONE. We note that your study design may include death of a regulated animal as a likely outcome or planned experimental endpoint. At this time, we request that you please report additional details in your Methods section regarding animal care and use for the survival study, as per our editorial guidelines (http://journals.plos.org/plosone/s/submission-guidelines#loc-humane-endpoints).      

For easy reference, we have attached a checklist that may be relevant for your submission. Please complete all items on the checklist at the following link:   http://journals.plos.org/plosone/s/file?id=bb1d/plos-one-humane-endpoints-checklist.docx         

Please upload the completed checklist as file type “Other” when resubmitting your manuscript. This document is for internal journal use only and will not be published if your article is accepted. We very much appreciate your attention to these requests and support of improved reporting standards in PLOS ONE submissions.

Reviewers' comments:

Reviewer's Responses to Questions

**Comments to the Author**

1. Is the manuscript technically sound, and do the data support the conclusions?

Reviewer #1: Yes

Reviewer #2: Yes

2. Has the statistical analysis been performed appropriately and rigorously? 

Reviewer #1: I Don't Know

Reviewer #2: I Don't Know

3. Have the authors made all data underlying the findings in their manuscript fully available?

Reviewer #1: Yes

Reviewer #2: Yes

4. Is the manuscript presented in an intelligible fashion and written in standard English?

Reviewer #1: No

Reviewer #2: No

5. Review Comments to the Author

Reviewer #1: Thanks for your hard efforts. Please check the following comments

1- The abstract contains a lot of abbreviations that need to be rewritten fully (GSI, HIS, DEGs, HPG).

2- In the introduction section, the part between lines 58-66 needs to be rewritten to organize the aims of the study (please make the aims more direct)

3- In lines 76-77, what do you mean by; three tanks each in control and 77 treatment groups? Does that mean 3 tanks for each group?

4- What is the vegetable oil that has been used in this study? Mention its source and commercial data.

5- In lines 79-80 the mention of Table 1 here is not correct, it should be Table 3 instead. The table 1 should be in the results section. In addition, in the result section the author used Table 2 in describing the results of the fertilization rate and GSI that were described in Table 1, not Table 2.

6- In line 81 how did you know that the best treatment among these, i.e., T2

7- In line 88, what is the meaning of PTO and PCO?

8- How did you analyze the data statistically

9- In lines 155 – 156 please replace Table 2 with Table 1.

10- In line 181 Table 3 with Table 2.

11- In line 185 please replace the symptoms w3 and w6 with ω3 and ω6.

12- The discussion section needs to be rewritten (please do not repeat the results in the discussion)

Reviewer #2: Title:

Please modify Title; You have not addressed the source of vegetable oil in Abstract, and M&M. Please add this to The Title, Abstract, and M&M.

And in the title add HPG following Hypothalamus Pituitary Gonadal axis

Please modify "Jitendra Kumar Sundaray and Janmejay Parhi should be considered as joint corresponding author" to Jitendra Kumar Sundaray and Janmejay Parhi are equally responsible as corresponding Author.

Abstract:

Line 31 Please specify the source of vegetable oil

Line 33 Please indicate the full form of abbreviated words when the first appear in the text, i.e. GSI and HSI.

Introduction:

Line 42 change gonad to gonads

Line 43 change muscle to muscles

Line 50 change like to including

Line 56 change Fish to fish

Line 64 change So to therefore

M&M

Line 112 please revise and edit

Line 118 as above

Results

OK

Discussion:

Line 183 Please replace "depict" with an appropriate verb. Depict means represent by a drawing, painting, or other art form.

Again, Please indicate the source of vegetable oil.

Line 184 change "Several studies earlier describe" to Several earlier studies showed..

Line 184 Several studies earlier describe the beneficial effects of dietary lipid on

ovarian development (29-32). In which species?

Line 185 please correct omega symbol

Line 186 vitamins

Line 186 change gave a better result to gave a better understanding

Line 191 please revise. The writing style needs professional check.

Line 221 please revise

Line 267 can hence what?

Line 273 it can not

6. PLOS authors have the option to publish the peer review history of their article (what does this mean?). If published, this will include your full peer review and any attached files.

Reviewer #1: No

Reviewer #2: **Yes: **Reza Masoumi, A/Prof of Animal Physiology

---

## [Author Response · Author response to Decision Letter 0]

12 Jul 2024

We have addressed the reviewer comments in response to reviewer file.

In addition to this the ethical consideration statements have been mentioned well as per plosone requirements.

Moreover the plos-one-humane-endpoints-checklist.docx has been uploaded along with the revised manuscript.

---

## [Decision Letter · Decision Letter 1]

9 Aug 2024

Effect of vegetable oil on ovarian steroidogenesis- A transcriptome approach to understand molecular mechanisms of hypothalamus pituitary and gonad axis (HPG) in Ompok bimaculatus

PONE-D-24-14273R1

Dear Dr. Jitendra Kumar,

We’re pleased to inform you that your manuscript has been judged scientifically suitable for publication and will be formally accepted for publication once it meets all outstanding technical requirements.

Kind regards,

Amel Mohamed El Asely

Academic Editor

PLOS ONE

Additional Editor Comments (optional):

Reviewers' comments:

Reviewer's Responses to Questions

**Comments to the Author**

1. If the authors have adequately addressed your comments raised in a previous round of review and you feel that this manuscript is now acceptable for publication, you may indicate that here to bypass the “Comments to the Author” section, enter your conflict of interest statement in the “Confidential to Editor” section, and submit your "Accept" recommendation.

Reviewer #1: All comments have been addressed

2. Is the manuscript technically sound, and do the data support the conclusions?

Reviewer #1: Yes

3. Has the statistical analysis been performed appropriately and rigorously? 

Reviewer #1: I Don't Know

4. Have the authors made all data underlying the findings in their manuscript fully available?

Reviewer #1: Yes

5. Is the manuscript presented in an intelligible fashion and written in standard English?

Reviewer #1: Yes

6. Review Comments to the Author

Reviewer #1: Thanks for the hard work. Your paper has a novel idea, good methodology, and well written. The reviewing items that I have asked are completely answered. Good luck

7. PLOS authors have the option to publish the peer review history of their article (what does this mean?). If published, this will include your full peer review and any attached files.

Reviewer #1: **Yes: **AHMED ELKHAWAGAH

---

## [Editor Report · Acceptance letter]

26 Sep 2024

PONE-D-24-14273R1 

PLOS ONE

Dear Dr. Sundaray, 

I'm pleased to inform you that your manuscript has been deemed suitable for publication in PLOS ONE. Congratulations! Your manuscript is now being handed over to our production team.

Kind regards, 

on behalf of

Prof. Amel Mohamed El Asely 

Academic Editor

PLOS ONE